# Single-Cell RNA Sequencing Reveals an Atlas of Meihua Pig Testis Cells

**DOI:** 10.3390/ani15050752

**Published:** 2025-03-05

**Authors:** Mao Zhang, Yiming Yan, Guoliang Peng, Shuang Gao, Hongyi Li, Yuan Li

**Affiliations:** College of Biology and Agriculture, Shaoguan University, Shaoguan 512005, China; zm18email@sgu.edu.cn (M.Z.); yimingyan0723@163.com (Y.Y.); pengguoliang168@163.com (G.P.); sunnie_scau@163.com (S.G.)

**Keywords:** single-cell transcriptome sequencing, Meihua pig, testis, TKTL1

## Abstract

We performed an unbiased transcriptomic study of spermatogenesis in neonatal and sexually mature Meihua pigs using 10× Genomics single-cell RNA, and identified three germ cell types and eight somatic cell types from 13,839 cells. Pseudo-timing analysis showed that myoid cells and stromal cells originated from common progenitors in Meihua pigs. Functional enrichment analysis revealed that the differentially expressed genes in testicular somatic cells and germ cells were enriched in their specific pathways, but some of the pathways were the same. Furthermore, we identified TKTL1 as a new marker gene for spermatogonia in pigs.

## 1. Introduction

The testis is an important organ in the male reproductive system of mammals, responsible for producing sperm and hormones, mainly composed of seminiferous tubules and testicular mesenchyme [1,2]. Spermatogenesis originates from the basement membrane of the seminiferous tubules in the testes and relies on spermatogonial stem cells (SSCs) [3]. Firstly, undifferentiated spermatogonia can self-renew to maintain numerical balance. Secondly, spermatogonia differentiate and further divide to form primary spermatocytes, which undergo two meiotic divisions starting from the primary oocyte to produce secondary spermatocytes and round sperm. Finally, sperm cells develop from round cells into highly specialized sperm, which are ultimately released into the lumen of the seminiferous tubules [4,5,6]. The entire process of spermatogenesis is complex but precise, orderly and efficient, in which germ cells (spermatogonia, spermatocytes, spermatids) and somatic cells (Sertoli cells, Leydig cells, Endothelial cells, etc.) play key roles in the processes of mitosis, meiosis, cell deformation and the formation of mature sperm [2,7,8,9,10]. However, there is heterogeneity between germ cells and somatic cells in the testes, which poses significant challenges in studying various cell types and gene expression at different developmental stages [11].

The proliferation and differentiation of spermatogonia are regulated by the microenvironment, composed of various surrounding cells, and the complexity and diversity of testicular cells make it challenging to study spermatogenesis, especially the proliferation and differentiation of spermatogonia.

In recent years, the rapid development of single-cell RNA sequencing (scRNA-seq) technology has provided us with powerful tools for studying gene expression and cellular changes during mammalian spermatogenesis and development. Through the sequencing of the RNA transcripts of individual cells and comparisons of them at gene expression levels, it is possible to determine which genes are expressed in different types of germ cells and reveal their different functions and regulatory mechanisms [12]. For example, scRNA-seq analysis of testicular cells identified new molecular markers of germ cells, which are closely related to the development and differentiation status of germ cells. In addition, testicular cells were grouped and the dynamic process and key regulatory factors of spermatogenesis were revealed [13,14,15,16,17]. In addition to the identification of molecular markers for germ cells, new markers for Sertoli cells have also been identified [18]. Through scRNA-seq analysis combined with lineage tracing technology, it was found that FOXC2+ and Eomes+ labeled cells in the subpopulation of spermatogonial stem cells represent quiescent spermatogonial stem cells, which is of great significance for reproduction and stem cell biology [19,20]. Research conducted on testicular development and spermatogenesis in humans [12,21], rodents [8,16,22,23,24] and primates [25,26] has expanded the understanding of the process of sperm development. With the expansion of research subjects, studies on the single-cell sequencing of testes in livestock such as sheep [11,27], dairy goats [28], buffalo [29], cattle [30] and yak [14], pigs [31,32,33] and horses [34] have also been carried out.

As an important agricultural economic animal, pigs play a crucial role in animal husbandry and are the main source of meat products. At the same time, pigs are also an important animal model in biomedical research [35]. The Meihua pig is representative of local pig breeds in Guangdong Province, with excellent qualities such as long service life, early sexual maturity, tolerance to roughage, strong adaptability, strong disease resistance and delicious meat. The fertility and sperm production ability of boars have important economic significance for the pig farming industry. Therefore, a deep understanding of the process of spermatogenesis in pigs, especially the developmental pattern of SSCs in pigs, is crucial for pork production, pig breeding and breed protection. However, research on the process of sperm production in Meihua pigs is still very limited. Conducting scRNA-seq on Meihua pig testicular tissue to construct a cellular atlas of the testis and identify novel markers for spermatogonia will provide valuable references for research on pig reproduction and spermatogonial biology. This will enhance our understanding of testicular development patterns and spermatogenesis across different pig breeds.

## 2. Materials and Methods

### 2.1. Testicular Sample Collection

One 1-week-old (W1) Meihua pig and one 6-month-old (M6) Meihua pig (Meihua Town, Shaoguan, Guangdong Province, China) were euthanized by the injection of sodium pentobarbital solution, and the testicular tissues were collected for histological tests and scRNA-seq. The collected testes were washed with Dulbecco’s phosphate-buffered saline (DPBS, Servicebio, Wuhan, China) and cut into 0.5 cm^3^ sized blocks, and then soaked in centrifuge tubes containing 2% (*v*/*v*) penicillin–streptomycin in Dulbecco’s modified eagle medium (DMEM, Servicebio, Wuhan, China). Ice cubes were placed around them to maintain low temperature, and they were transported back to the laboratory for cell separation within 24 h.

### 2.2. Histological Observation

The collected testes were cut into appropriately sized blocks and placed in an animal testicular tissue fixative (Servicebio, Wuhan, China). After fixation for 24 h, paraffin sections were made and stained with hematoxylin and eosin (HE).

### 2.3. Preparation of Single-Cell Suspension

The isolation and purification of single-cell testicular tissue were conducted using a two-step enzymatic method. Briefly, the testicular tissue was cut into pieces using ophthalmic scissors, and 1 mg/mL of collagen type IV (Gibco, Waltham, MA, USA) was incubated at 37 °C for 20 min with 5 min intervals for shaking. After washing with pre-cooled DPBS (Gibco, Waltham, MA, USA), 0.25% trypsin (Gibco, Waltham, MA, USA) and 0.25 mg/mL DNase I (Sigma-Aldrich, St. Louis, MO, USA) were added and cultured at 37 °C for 10 min. Digestion was terminated by adding DMEM (Gibco, Waltham, MA, USA) containing 10% FBS (Gibco, USA), and testicular single cells were obtained after filtration through a 40 μm cell sieve. The filtrate was collected and centrifuged at 300× *g* at 4 °C for 5 min and washed with pre-cooled DPBS. A hemocytometer (Thermo Fisher Scientific, Waltham, MA, USA) was used to evaluate cell viability rates, and the concentration of the single-cell suspension was calculated using Countstar, with cell viability > 80% and an adjusted cell concentration of 700–1200 cells/μL for subsequent single-cell sequencing.

### 2.4. Single-Cell Sequencing

The cell suspension was loaded into Chromium microfluidic chips with 3′v3 chemistry and barcoded with a 10× Chromium Controller (10× Genomics). RNA from the barcoded cells was subsequently reverse-transcribed and sequencing libraries constructed with reagents from a Chromium Single Cell 3′v3 reagent kit (10× Genomics, Pleasanton, CA, USA) according to the manufacturer’s instructions. Library construction and single-cell sequencing were completed using the PE150 mode of the Illumina Novoseq 6000 sequencing platform at Novogene Laboratories (Guangzhou, China).

### 2.5. Quality Control, Mapping and Clustering Analysis

The sequenced reads were stored in FASTQ format, and fastq software (version 0.20.0) was used to perform basic statistics on the quality of raw reads. We used 10× Genomic software Cell Ranger (version 7.1.0) to align the previously generated fastq sequencing data with Scrofa11.1 for cell and UMI counting, generating a cell gene expression matrix. Secondary filtration performed by Seurat (version 4.3.0). A gene with expression in more than 3 cells was considered expressed, and each cell was required to have at least 200 expressed genes. Some of the foreign cells were filtered out. Based on the filtered cell gene expression matrix, Seurat standardized the data and used principal component analysis (PCA) to reduce their dimensionality. Then, tSNE/UMAP was used to cluster the data and obtain different cell subgroups. Finally, the definition of cell types for each cell cluster was based on reported marker genes.

### 2.6. Cell Trajectory Analysis

Pseudo-time analysis, also known as cell trajectory analysis, can infer the differentiation trajectory or evolution process of cell subtypes during development based on the changes in gene expression levels of different cell subgroups over time. In this study, monocle 2 (version 2.26.0) was used in the pseudo-time analysis of germ cells and somatic cells.

### 2.7. Differentially Expressed Gene (DEG) and Functional Enrichment Analysis

By comparing a subgroup of the sample with all other subgroups, differential genes between the cells in this subgroup and those in other subgroups can be obtained, providing a basis for the further screening of marker genes. Gene Ontology (GO) enrichment analysis of marker genes was implemented by the clusterProfiler (version 3.14.0) R package, in which gene length bias was corrected. GO terms with a corrected P-value less than 0.05 were considered significantly enriched by the marker gene. We used the clusterProfiler R package to test the statistical enrichment of marker genes in KEGG pathways.

### 2.8. Immunohistochemistry and Immunofluorescence Analysis

Immunohistochemical detection of spermatogonia in testicular tissue was performed using TKTL1 antibody. Paraffin sections were deparaffinized and antigen-repaired. After sealing, rabbit-derived antibody TKTL1 (sabbiotech, College Park, MD, USA, diluted 1:100) was incubated overnight at 4 °C. Then, HRP-labeled goat anti-rabbit secondary antibody (Servicebio, Wuhan, China, 1:200) was incubated at room temperature for 50 min, followed by DAB staining and hematoxylin counterstaining of the cell nucleus. Double immunofluorescence detection was performed using UCHL1 and TKTL1. Mouse-derived primary antibody UCHL1 (proteintech, Wuhan, China, 1:4000) was added and incubated overnight at 4 °C, followed by the corresponding HRP-labeled secondary antibody at room temperature for 50 min. After washing with PBS, the corresponding TSA dye (iF555 Tyramide, Servicebio, Wuhan, China) was added and incubated at room temperature in the dark for 10 min. After washing, microwave treatment was performed. Then, the second primary antibody TKTL1 (sabbiotech, USA, 1:500) was added and incubated overnight at 4 °C, followed by the corresponding HRP-labeled secondary antibody at room temperature for 50 min. After washing, the corresponding TSA dye (IF488 Tyramide, Servicebio, Wuhan, China) was added and incubated at room temperature in the dark for 10 min, followed by DAPI (Servicebio, Wuhan, China) counterstaining of the cell nucleus.

## 3. Results

### 3.1. Histological Observation of Testicular Cell Types in Neonatal and Sexually Mature Meihua Pigs 

To observe the different cell types and ensure the samples were appropriate for sequencing, different somatic and germ cells from two collected testicle tissues were observed. The seminiferous tubule lumen of the 1-week-old Meihua pig was substantial, and there were spermatogonia and Sertoli cells in the lumen (Figure 1A), while the seminiferous tubule lumen of the 6-month-old Meihua pig testis was formed and there was obvious spermatogenesis (Figure 1B).

### 3.2. Subset Analysis and Cell Type Identification of Testicular Cells in Meihua Pigs

After dissociation, the single-cell suspensions from the W1 and M6 tissues were obtained and the cell viability was 93.9% and 88.5%, respectively. A total of 14,663 cells from W1 (8000 cells) and M6 (6663 cells) were sequenced, and 325.5 Mb and 322.7 Mb reads were generated from the sequencing library, respectively. The reads of each cell were 41,662 for W1 and 49,550 for M6, and the median of gene detection was 1498 (W1) and 2946 (M6), respectively. More information is available in Appendix A. Finally, 13,839 filtered cells from W1 (7549 cells) and M6 (6290 cells) were integrated and divided into 27 cell clusters via t-SNE (Appendix A) and UMAP clustering (Figure 2A,B). The cell clusters were first classified into germ cells and somatic cells based on the expression of DDX4 and VIM as previously reported [26,31], which were further identified according to the known cell marker genes (Figure 2C and Appendix A). The cells in clusters 14 and 24 expressed spermatogonia marker genes UCHL1, ETV5 and PLD6. The cells in clusters 2, 3, 15 and 16 solely expressed spermatocyte marker genes NME8 and SYCP1. Spermatid cells consisted of clusters 5, 6, 8 and 11 and specifically expressed the TPPP2 and PRM1 marker genes. Clusters 0, 4, 9, 12 and 19 consisted of cells expressing Sertoli cell marker genes AMH, GATA4 and Sox9. Cluster 25 expressed myoid cell marker genes MYH11 and ACTA2. Clusters 1 and 21 were identified as myoid/stromal cells, as cells in these clusters specifically expressed ACTA2 and COL1A1l. Cluster 20 expressed stromal cell marker genes COL1A1 and COL3A1. Clusters 7 and 23 expressed Leydig cell marker genes CYP11A1 and CYP17A1. The cells in cluster 14 expressed terminal cell marker genes VWF and PECAM1. The cells in cluster 22 expressed the erythroblast marker gene HBM. The cells in clusters 13 and 18 expressed macrophage markers CTSS and TYROBP, and the cells in cluster 10 expressed T cell marker CD3D. This study successfully identified the main cell types in Meihua pig spermatogenesis (Figure 2D). Sertoli cells account for the highest proportion in W1, with 54.06%, while the main cells in M6 were spermatocytes (38.71%) and spermatids (33.53%) (Figure 2E, Appendix A).

### 3.3. Cell Trajectory Analysis

Cluster analysis with dimensionality reduction indicated that myoid cells were located in clusters 1 and 25, while stromal cells were in clusters 1, 20 and 21. The myoid cell marker gene TCTA2 and stromal cell marker gene COL1A1 were expressed in clusters 1, 20, 21 and 25 (Figure 3A,B). Myoid cells and stromal cells have been reported to develop from the same progenitor cell (marked by TOP2A) in Bama pigs [36]. Thus, a pseudo-time analysis of clusters 1, 20, 21 and 25 was performed to investigate whether this was also the case for Meihua pigs. The results of the analysis showed that myoid cells and stromal cells differentiated from a common progenitor cell, and we also found that the TOP2A gene was specifically highly expressed in cluster 21, which was shown earlier in terms of pseudo-time (Figure 3C,D).

### 3.4. Functional Enrichment Analysis of Testicular Somatic Cells in Meihua Pigs

GO-term functional enrichment and KEGG analyses of DEGs in testicular somatic cells were performed to explore the distinct functions of the identified somatic cells (Appendix A). GO analysis indicated that DEGs of Sertoli cells were mainly enriched in the translation, intracellular protein transport, protein deubiquitination and ubiquitin-dependent protein catabolic process of the biological process category. They were also enriched in the molecular function category, including the terms structural constituent of ribosome, translation, cysteine-type deubiquitinase activity and so on. In the cellular component category, they were enriched in the terms ribosome and proteasome core complex (Figure 4A). Meanwhile, in myoid/stromal cells, DEGs mainly involved in the biological process were similar to those in Sertoli cells, those in the molecular function category were mainly involved in RNA binding, structural constituent of ribosome and protein kinase activity, and those in the cellular component category were involved in nucleus and endoplasmic reticulum (Figure 4C). Additionally, KEGG analysis revealed that DEGs of Sertoli cells were enriched in the pathways of Ribosome, Parkinson disease, Oxidative phosphorylation, Thermogenesis, Retrograde endocannabinoid signaling and Protein processing in endoplasmic reticulum (Figure 4B). Except for the same pathways of Ribosome, Oxidative phosphorylation, Retrograde endocannabinoid signaling and Protein processing in endoplasmic reticulum, the DEGs in myoid/stromal cells also participate in the Longevity regulating pathway, Cellular senescence and Insulin signaling pathway (Figure 4D). Moreover, other somatic cells such as Leydig cells, T cells/macrophages and endothelial cells have their own functions (Appendix A).

### 3.5. Functional Enrichment Analysis of Testicular Germ Cells

To investigate the function of germ cells at different stages, we conducted GO and KEGG analyses on germ cells in different states (Appendix A) and found that DEGs involved in the process of spermatogenesis from spermatogonia to spermatocytes were all related to the GO terms of mitochondrion, protein folding, microtubule-based movement, ATP hydrolysis activity and translation initiation factor activity (Figure 5A). Similarly, three germ cells were all enriched in pathways associated with protein synthesis and energy generation, such as Ribosome, Protein processing in endoplasmic reticulum, Oxidative phosphorylation and Spliceosome. Interestingly, the first three of the listed pathways were the same as those for somatic cells (Appendix A). Moreover, the DEGs of spermatogonia were also enriched in pathways related to mitosis, such as Cell cycle. While the DEGs of spermatocytes were also enriched in Cell cycle, Autophagy, mTOR signaling pathway and Mitophagy. In spermatids, the DEGs were also enriched in Mitophagy and Citrate cycle (Figure 5B and Appendix A).

### 3.6. Re-Clustering of Spermatogonia and Screening of New Markers

The spermatogonia (clusters 14 and 24) were further divided into six cell subgroups (Figure 6A), and the clustering results showed that the spermatogonia from W1 were distributed in sub-clusters 0 and 5 (Figure 6B). The specific expression marker gene UCHL1 can be further defined as undifferentiated spermatogonia (Figure 6C). Further analysis revealed that the TKTL1 gene was specifically highly expressed in these two cell populations (Figure 6D), suggesting that TKTL1 may serve as a marker gene for undifferentiated germ cells in Meihua pigs. To further verify the role of TKTL1 as a marker gene, immunohistochemical identification was performed on W1 and M6 testicular tissue. The results showed that TKTL1 was localized in spermatogonia in the seminiferous tubules (Figure 6E). Moreover, double-antibody immunofluorescence analysis showed that TKTL1 and UCHL1 were both localized in spermatogonia and highly overlapped in W1 testicular tissue (Figure 6F).

## 4. Discussion

The normal development of testicles and the continuous occurrence of sperm are important to maintain fertility for male animals, which require intricate interaction between testicular germ cells and somatic cells [37,38]. Consequently, it is of great significance to study the characteristics and functions of various testicular cells. The development and differentiation of germ cells are the main processes involved in spermatogenesis. However, the developmental kinetics of spermatogenesis are highly complex, and it is difficult to distinguish testicular cell types using traditional methods, which can only identify certain types. ScRNA-seq has become a powerful tool for solving such challenges effectively [39], with uses in identifying cell subpopulations, drawing cell maps, and the parallel analysis of various testicular cell states in the testes.

In this study, we performed scRNA-seq on 13,839 testicular cells of Meihua pigs from the neonatal and sexual maturity stages, and obtained 27 cell clusters. Based on reported marker genes, these clusters were divided into three germ cell types and eight somatic cell types, constructing a testicular cell landscape of the Meihua pig and providing valuable information for the further study of male reproduction in local pigs.

The testicular tissue is composed of seminiferous tubules, which are made up of Sertoli cells and germ cells, surrounded by peritubular myoid cells, with the interstitium located between the tubules [40]. Testicular somatic cells regulate spermatogenesis by secreting cytokines and modulating specific signaling pathways that control the balance between SSC self-renewal and differentiation [41]. Sertoli cells, which are located at the basilar membrane of seminiferous tubules, can provide structural support for SSCs and supply numerous critical cytokines and an extracellular matrix for SSC development. Among them, glial cell-derived neurotrophic factor (GDNF) participates in regulating the proliferation and differentiation of SSCs, while FGF2, WNT6 and WNT5a are involved in maintaining the renewal of spermatogonia [42,43]. Leydig cells are the major steroidogenic cell population in the testicular interstitium, and they are best known for producing testosterone, which is essential for the maintenance of spermatogenesis [44,45]. Myoid cells regulate the transport of sperm and luminal fluid and secrete growth factors and extracellular matrix components to maintain the niche of SSCs. For example, GDNF, produced by peritubular myoid cells, plays an important role in the development of undifferentiated spermatogonia in vivo [46]. Leydig cells and myoid cells have been reported to share common progenitors in humans [21], goats [47], yaks [14] and Hezuo pigs [33] using pseudo-time analysis. Additionally, Wang et al. [36] further studied the specification of the interstitial cell lineage in Bama pigs and found that myoid cells and stromal cells develop from a common progenitor cell. To reveal whether the above-mentioned phenomenon also exists in Meihua pigs, we conducted pseudo-time analysis and found that myoid cells and stromal cells originated from common progenitor cells, which was consistent with the result reported by Wang. Stromal cells are a large group of cells, with a relatively significant proportion, while myoid cells only occupy a small proportion after sexual maturity.

In this study, we essentially identified the previously reported somatic cell types such as Sertoli cells, Leydig cells, myoid cells, stromal cells, endothelial cells, macrophages and T cells, which account for the majority of the sequenced cells. The DEGs in each cell type and their enriched GO terms and pathways were also identified. It is worth noting that functional enrichment analyses of DEGs in testicular somatic cells have been undertaken in many species [11,13,14,31,33,38,47]. In Guangzhong pigs, it was reported that DEGs in myoid cells participated in growth factor response, extracellular matrix organization and the regulation of cell adhesion. DEGs in Leydig cells were mainly enriched in the metabolic processes of monocarboxylic acids and steroids, and DEGs in Sertoli cells were involved in the regulation of cell death and exocytosis [31]. Meanwhile, in Hezuo pigs, DEGs in Leydig cells and myoid cells were mainly enriched in the organic substance metabolic process, intracellular organelle part and basement membrane, and were mainly involved in the PI3K-Akt, vascular smooth muscle contraction and steroid biosynthesis pathways. DEGs in Sertoli cells were primarily associated with GO terms participating in the organic substance metabolic process, cell part and protein binding, and enriched in the thyroid hormone as well as in Wnt signaling pathways [13,33]. In our study, functional enrichment analysis of the DEGs in Meihua pigs showed that genes in somatic cells were enriched in mitochondrion, protein synthesis and activity-related terms with similar functions to pathways, such as Oxidative phosphorylation, Ribosome, Protein processing in endoplasmic reticulum and Autophagy. Notwithstanding the functional enrichment of DEGs differing among various breeds and ages, the main functions were all associated with spermatogenesis and testicular development, suggesting that the roles of somatic cells in different breeds may vary at different physiological stages.

As pigs are an important economic animal, the development mechanism of their germ cells is of great significance to husbandry. Through the performance of scRNA-seq on pig testicles, the gene expression profiles and transcriptional regulatory networks of germ cells, such as spermatogonia, spermatocytes and spermatids, could be deeply analyzed and the key genes and signaling pathways of these cells during development could be revealed, which provided a new visual angle from which to understand the proliferation, differentiation and apoptosis of germ cells. In this study, we identified three groups of germ cells, including spermatogonia, spermatocytes and spermatids, based on the reported marker genes (UCHL1, PLD6, NME8, SYCP1, TPPP2 and PRM1) [31,32,48]. What a pity that we failed to further identify the spermatogonial subpopulation since classical marker genes of spermatogonial differentiation such as KIT and STRA8 were not detected, which may be due to the small proportion of differentiated spermatogonia in Meihua pigs. When defining cells, we identified many specifically expressed genes that have not been defined or reported in various cell types, such as ENSSSCG00000052117, ENSSSCG00000054196, ENSSSCG00000034743 and ENSSSCG00000028425. These genes may play a critical role in spermatogenesis in Meihua pigs, suggesting that there may be a quantity of specific regulatory genes in Meihua pigs, which needs further annotation and exploration.

Transketolase like 1 (TKTL1) is a key regulatory enzyme in the pentose phosphate pathway (PPP) and plays an important role in energy synthesis [49,50]. Yu et al. performed scRNA-seq on the testes of preadolescent dairy goats and found that TKTL1 expression was highest in the testis tissue, and the protein was constantly expressed throughout the maturation of germ cells and localized in the spermatogonia [47]. Subsequently, Wang et al. defined the TKTL1 gene as a marker gene of yak spermatogonia [14]. However, whether TKTL1 can be used as a marker gene of porcine spermatogonia has not been verified. We re-clustered the spermatogonia and found that cells expressing TKTL1 were derived from the spermatogonial cells of neonatal Meihua pigs, with the same applying to cells expressing UCHL1, and therefore speculated that TKTL1 might be a marker gene of undifferentiated spermatogonia. We then localized TKTL1 to spermatogonia by immunohistochemistry, and immunofluorescence analysis showed high overlap between UCHL1-expressing cells and TKTL1-expressing cells, indicating that TKTL1 is a marker gene for spermatogonia in Meishan pigs. TKTL1 may serve as a novel marker for isolating porcine spermatogonia for in vitro studies. Additionally, it would be interesting to study the role of TKTL1 in the proliferation and differentiation of porcine spermatogonia.

## 5. Conclusions

We generated an expression atlas of 13,839 cells from the testes of neonatal and sexually mature Meihua pigs using scRNA-seq. We confirmed 23 marker genes for cell type classifications and identified 11 testicular cell types in Meihua pigs. We also identified TKTL1 as a new marker gene for Meihua pig spermatogonia, which might be a potential new marker gene in porcine spermatogonia. In conclusion, we characterized the transcriptomic landscape of Meihua pig spermatogenesis through comprehensive analysis and identified a new marker gene for spermatogonia. The datasets and results of this study are expected to provide a valuable theoretical basis for the investigation of male reproduction, genetic improvement and conservation in Meihua pigs.

## Figures and Tables

**Figure 1 animals-15-00752-f001:**
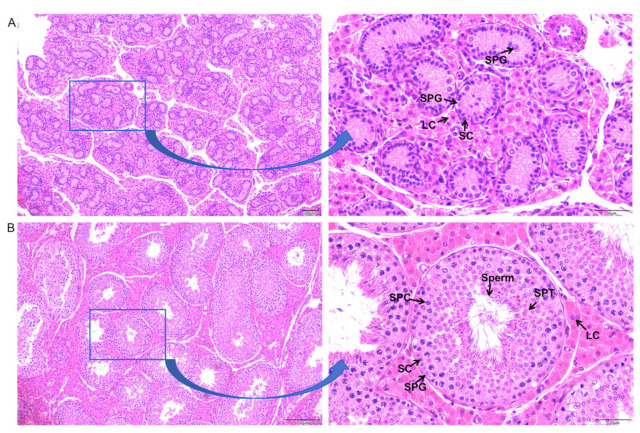
Histological observation of Meihua pig testicular tissue sections. (**A**) 1 week old. (**B**) 6 months old. SPG: spermatogonia; SPC: spermatocyte; SPT: spermatid; SC: Sertoli cell; LC: Leydig cell. Left bar = 200 µm; right bar = 50 µm.

**Figure 2 animals-15-00752-f002:**
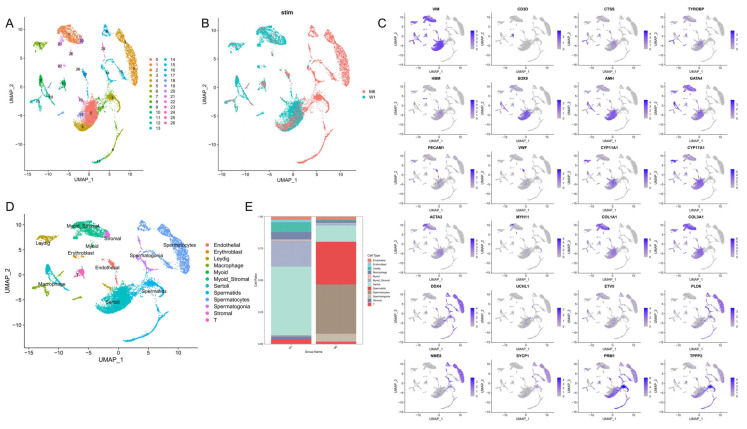
Classification of single-cell subpopulations and their visualization. (**A**) UMAP clustering analysis of all testicular cells. (**B**) Cell clusters in UMAP according to groups, with cells colored based on age. (**C**) The expression pattern of the selected marker genes on the UMAP map. (**D**) UMAP plots of testicular cell clusters defined by scRNA-seq analysis; different colors represent different cell types. (**E**) The percentage of cells in each of the 11 clusters in two samples (*Y*-axis). Clusters are distinguished by color.

**Figure 3 animals-15-00752-f003:**
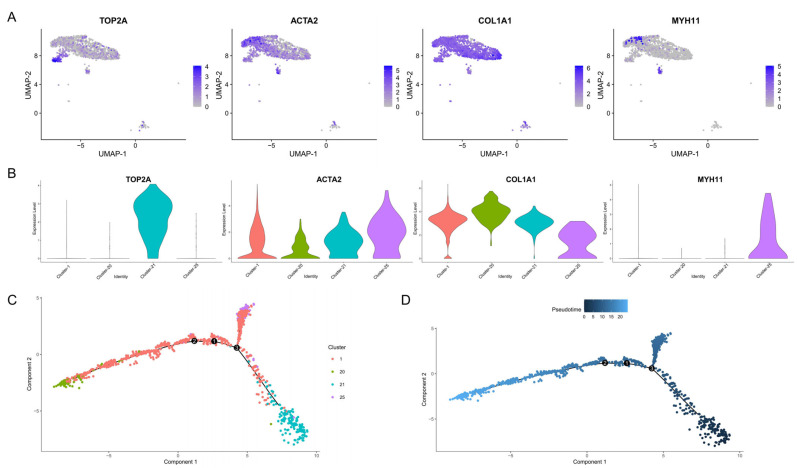
Pseudo-time analysis of myoid cells and stromal cells. (**A**) Marker genes of different cells with UMAP plot. (**B**) Violin plots of different cell type-specific gene expression patterns across different clusters. (**C**,**D**) The pseudo-time trajectory of the myoid cells and stromal cells, colored according to their predicted cluster (**C**) and pseudo-time. Cells with a darker color are in the front position of the pseudo-time, and cells with a lighter color are in the back position of the pseudo-time (**D**). The numbers in the figure represent the nodes of the pseudo-time trajectory.

**Figure 4 animals-15-00752-f004:**
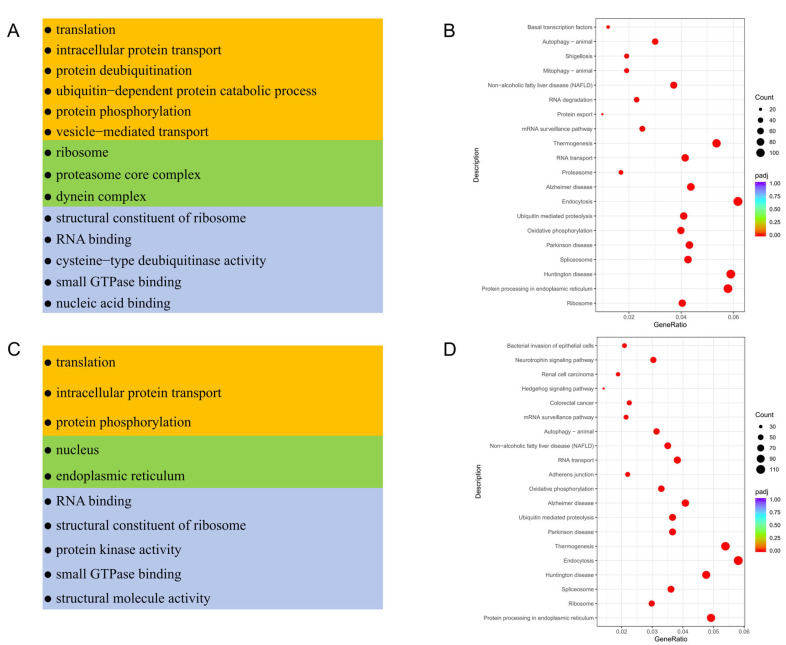
Functional enrichment analysis of different types of testicular somatic cells. (**A**) GO analysis of DEGs in Sertoli cells. (**B**) KEGG analysis of DEGs in Sertoli cells. (**C**) GO analysis of DEGs in myoid/stromal cells. (**D**) KEGG analysis of DEGs in myoid/stromal cells.

**Figure 5 animals-15-00752-f005:**
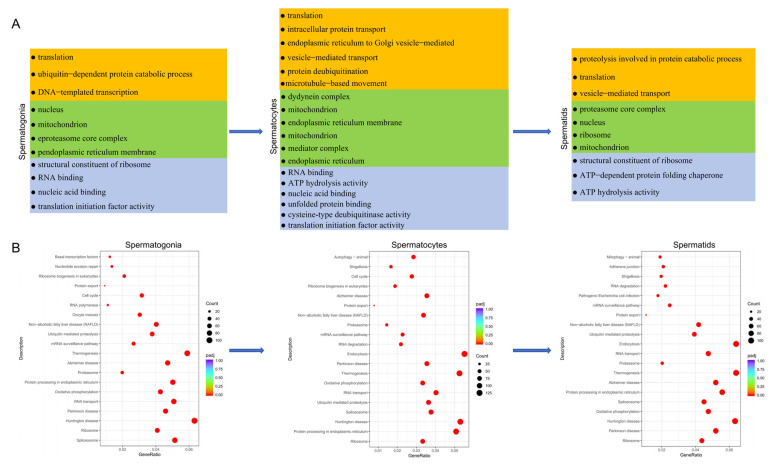
Functional enrichment analysis of testicular germ cell types. (**A**) GO analysis of DEGs in germ cell types. (**B**) KEGG analysis of DEGs in germ cell types.

**Figure 6 animals-15-00752-f006:**
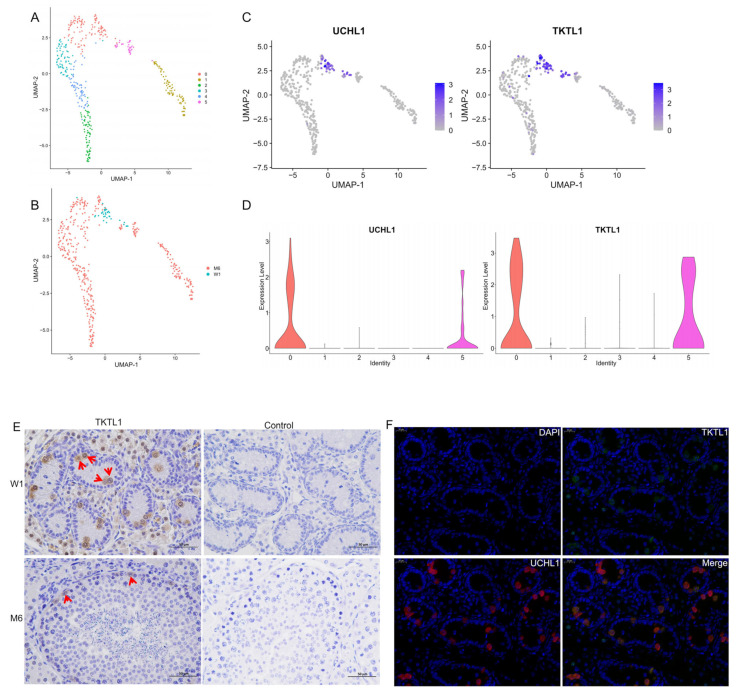
Expression and distribution of potential marker gene. (**A**,**B**) Re-clustering of spermatogonia, colored by assignment to six sub-clusters (**A**) and two samples (**B**). (**C**) Expression of UCHL1 and TKTL1 in Meihua pig spermatogonia. (**D**) Violin plots of UCHL1 and TKTL1 expression across six sub-clusters. (**E**) Immunohistochemistry analysis of location of TKTL1 in Meihua pig testis. The red arrows indicated the TKTL1-positive cells. Bar = 50 µm. (**F**) Immunofluorescence analysis of location of TKTL1 and UCHL1 in Meihua pig testis. Bar = 20 µm.

## Data Availability

The sequencing data have been uploaded to the National Genomics Data Center database under Bioproject No. PRJCA034729 (https://ngdc.cncb.ac.cn/bioproject, accessed on 8 January 2025).

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
