# Peer review of "Single-Cell RNA Sequencing Reveals an Atlas of Meihua Pig Testis Cells"

_animals, 2025, doi:10.3390/ani15050752_

Round 1
Reviewer 1 Report
Comments and Suggestions for Authors
Why were not other pig breeds or mammals included for comparison? Would discussing the potential applications of TKTL1 as a marker for undifferentiated spermatogonia in reproductive biology or genetics add value to your study? What about summarizing key pathways in a comparative table between somatic and germ cells? Maybe could improve the clarity of the KEGG and GO analyses. I have a last quetion about the origin of myoid and stromal cells. Were they validated using any lineage tracing?
Author Response
Comments 1: Why were not other pig breeds or mammals included for comparison?
Response 1: Our study is focused on mapping the transcriptional landscape of testicular cells in Meihua pigs through single-cell sequencing and identifying novel marker genes for spermatogonia. While we have only analyzed the sequencing data obtained from Meihua pigs, we have referenced studies on other pig breeds and domestic animals, and compared our findings with existing research. It is a good idea to incorporate and compare sequencing data from other breeds or species, thank you very much.
Comments 2: Would discussing the potential applications of TKTL1 as a marker for undifferentiated spermatogonia in reproductive biology or genetics add value to your study?
Response 2: Yes, we have already added this discussion in the relevant section. Additionally, we will further investigate the role of TKTL1 in the proliferation and differentiation of porcine spermatogonia, thank you very much.
Comments 3: What about summarizing key pathways in a comparative table between somatic and germ cells?Maybe could improve the clarity of the KEGG and GO analyses.
Response 3: Great suggestion, Since the content of the table somewhat overlaps with the figure, the summary table has been placed in the supplementary materials (Table S5), thank you very much.
Comments 4: Were myoid and stromal cells validated using any lineage tracing?
Response 4: This is an excellent question. Lineage tracing can verify whether cells originate from a common progenitor. However, due to the significant challenges associated with this validation, we did not perform lineage tracing in this study. In the next steps, we plan to enhance our research by sequencing samples from different developmental stages and conducting additional cell-based experiments to further validate our findings, thank you very much.
Reviewer 2 Report
Comments and Suggestions for Authors
The study utilized a powerful technology (sc-RNAseq), and the presentation of the results is adequate.
Major concerns:
1. General criticism – The scRNAseq results were not validated, and this issue was not even discussed.
2. Introduction – There is no clear hypothesis, and the aim of the study appears to be the description of transcript patterns in the testicles of a local pig breed. One may question whether studying such patterns in a single breed is justified, given the existence of hundreds of pig breeds.
3. Materials
Line 106 – It should be explicitly stated whether two animals were studied. Additionally, were these animals related?
Line 108 – The phrase "into 0.5 cm sized blocks" is unclear. Does it refer to the volume of the blocks (0.5 cm³)?
4.Results
Line 303 – The values should be corrected to 325.5 Mb and 322.7 Mb.
5. Discussion – Recently, four articles on scRNAseq in porcine testicles were published, but only one of them (Tang et al., Biomolecules 2024) was discussed and cited.
The following three studies were not mentioned:
- Yan et al., Single-Cell RNA Sequencing Reveals an Atlas of Hezuo Pig Testis Cells. Int J Mol Sci. 2024.
- Wang et al., Single-cell transcriptomic and cross-species comparison analyses reveal distinct molecular changes of porcine testes during puberty. Commun Biol. 2024.
- Yan et al., Transcriptional Profiling of Testis Development in Pre-Sexually-Mature Hezuo Pig. Curr Issues Mol Biol. 2024.
6. Conclusion – The conclusion should focus on comparing the obtained results with the above-mentioned studies on scRNAseq in porcine testicles.
Author Response
Comments 1: General criticism – The scRNAseq results were not validated, and this issue was not even discussed.
Response 1: The sequencing results were validated by immunohistochemistry and immunofluorescence for new marker genes, which were also discussed in the discussion section,thank you very much.
Comments 2: Introduction – There is no clear hypothesis, and the aim of the study appears to be the description of transcript patterns in the testicles of a local pig breed. One may question whether studying such patterns in a single breed is justified, given the existence of hundreds of pig breeds.
Response 2: We have revised the Introduction and added statement of the objectives and hypothesis,thank you very much.
Comments 3: Materials
Line 106 – It should be explicitly stated whether two animals were studied. Additionally, were these animals related?
Response : Yes, two animals were studied, one was a newborn, and the other was sexually mature. We have added specific descriptions in the Materials and Methods section2.1, thank you very much.
Line 108 – The phrase "into 0.5 cm sized blocks" is unclear. Does it refer to the volume of the blocks (0.5 cm³)?
Response: We have revised the data to 0.5 cm³, thank you very much.
Comments 4: Results
Line 303 – The values should be corrected to 325.5 Mb and 322.7 Mb.
Response 4: Is it line 203? We have corrcted the values,thank you very much.
Comments 5: Discussion – Recently, four articles on scRNAseq in porcine testicles were published, but only one of them (Tang et al., Biomolecules 2024) was discussed and cited.
The following three studies were not mentioned:
- Yan et al., Single-Cell RNA Sequencing Reveals an Atlas of Hezuo Pig Testis Cells. Int J Mol Sci. 2024.
- Wang et al., Single-cell transcriptomic and cross-species comparison analyses reveal distinct molecular changes of porcine testes during puberty. Commun Biol. 2024.
- Yan et al., Transcriptional Profiling of Testis Development in Pre-Sexually-Mature Hezuo Pig. Curr Issues Mol Biol. 2024.
Response 5: We have already cited three of them in the original manuscript,they are references 32, 33, and 36, respectively,thank you very much.
Comments 6: Conclusion – The conclusion should focus on comparing the obtained results with the above-mentioned studies on scRNAseq in porcine testicles.
Response 6: Compare with other poricne testicles, we identified TKTL1 as a new marker gene for spermatogonia, and we have revised the conclusion,thank you very much.
Reviewer 3 Report
Comments and Suggestions for Authors
Comments and Suggestions for Authors
In this study, the authors performed an unbiased transcriptomic study of spermatogenesis in neonatal and sexually mature Meihua pigs using 10×Genomics Single-cell RNA, and conducted a series of analyses on the sequencing data, with robust data and innovative findings, expecting to provide valuable theoretical basis for the spermatogenesis of Meihua pig. It is a topic of interest to the researchers in the related areas but the paper needs some modification before publication.
# Abstract
Lines 16, 20 have grammatical errors. Please read the whole text carefully and make corrections. “regulated – is regulated”, “were – was”.
# Introduction
Line 61, the position of the abbreviations of “scRNA-seq” is incorrect.
# Results
①The font size of the titles on the vertical axes in figures 2 to 5 is generally too small. It is recommended to enlarge them appropriately. Figure 1. It is suggested that the bar numbers were marked in the pictures.
②The authors need to be aware of the punctuation mark, for example, line 263, 265 and 266.
③Line 317 to 320. The authors identified TKTL1 as the candidate marker gene in spermatogonia (Figure 6E and 6F), which was an important result for this paper. But the Figure 6E and Figure 6F are not presented in the article.
④The authors need to unify the regular and italic fonts of gene name, for example, line 319 and 326.
# References
Reference formatting should be consistent (e.g., some journal names are not abbreviated, such as “International Journal of Molecular Sciences” lines 458, 478, 519).
Author Response
Comments 1: #Abstract
Lines 16, 20 have grammatical errors. Please read the whole text carefully and make corrections. “regulated – is regulated”, “were – was”.
Response 1: We have corrected the grammatical errors,thank you very much.
Comments 2: #Introduction
Line 61, the position of the abbreviations of “scRNA-seq” is incorrect.
Response 2: We have corrected the position,thank you very much.
Comments 3: #Results
①The font size of the titles on the vertical axes in figures 2 to 5 is generally too small. It is recommended to enlarge them appropriately. Figure 1. It is suggested that the bar numbers were marked in the pictures.
Response: We have makred the bar numbers in the pictures,thank you very much.
②The authors need to be aware of the punctuation mark, for example, line 263, 265 and 266.
Response: We have corrected the the punctuation mark,thank you very much.
③Line 317 to 320. The authors identified TKTL1 as the candidate marker gene in spermatogonia (Figure 6E and 6F), which was an important result for this paper. But the Figure 6E and Figure 6F are not presented in the article.
Response: This was an oversight on our part, we have added the Figure 6E and 6F in the article,thank you very much.
④The authors need to unify the regular and italic fonts of gene name, for example, line 319 and 326.
Response: We have unified all the gene name to regular fonts,thank you very much.
Comments 4: #Reference
Reference formatting should be consistent (e.g., some journal names are not abbreviated, such as “International Journal of Molecular Sciences” lines 458, 478, 519).
Response 4: The format of the references has been modified,thank you very much.
Reviewer 4 Report
Comments and Suggestions for Authors
Comments on manuscript Animals-3456085-peer-review-v1.
General comments.
The manuscript titled “Single-Cell RNA Sequencing Reveals an Atlas of Meihua Pig Testis Cells” presents novel findings that contribute to enhancing understanding of spermatogenesis and testicular development in Meihua pigs through transcriptomic analysis of Meihua pig testis using single-cell RNA sequencing. However, before it can be considered for publication, some revisions are required, as many areas need addressing. The specific elements need correction and are detailed below. The detailed feedback aims to enhance the clarity, accuracy, and overall quality of the manuscript for publication.
Specific comments.
Line 18: The phrase "To better understand the pattern of various testicular cells in spermatogenesis" serves as a vague and poorly defined objective. The authors need to provide a clear and specific objective.
Lines 89 to 97: There is no need to include details about the materials and methods in this section, nor should findings or conclusions be presented here as it is not the appropriate location. The statement "To demonstrate the… and preservation of the race" should be removed. Instead, it is essential to clearly state the research objective and hypothesis.
Lines 106 to 107. It is not clear how the testicles were obtained. Were the pigs castrated or were they killed and then got their testicles back? How many one-week-old and six-month-old pigs were used in the experiment?
Materials and methods. Throughout the entire section, the authors must include a complete description of every one of the software, products, equipment, or commercial reagents used, to ensure the repeatability of the experiment. Please include information as follows: Trade name of the product, equipment, or reagent; name of the manufacturing company, place of manufacture.
Results: The order of the sections is incorrect, which renders the manuscript confusing and difficult to follow. The authors should reorder the sections to precisely match the sequence presented in the Materials and Methods section.
Lines 186 to 189: Please eliminate the redundant phrase “To observe the... cells were observed,” as this has already been described in the Materials and Methods.
Lines 259 to 260: Please remove the unnecessary statement “Somatic cells provide... and spermatogenesis.”
Lines 289 to 290: Please delete the redundant text “Spermatogenesis is a... differentiation.”
Lines 342 to 343: Please remove the statement “China has abundant... different regions,” since the study did not assess differences between pig breeds, making this content irrelevant.
Lines 378 to 379. Please remove the text “The results may vary depending on the species, breed, and age”. The study did not evaluate differences between breeds of pigs, so the authors are assuming untested results.
Conclusions (pp. 429-435): The authors present a thorough summary of the manuscript; however, the overall organization and direction are lacking. The manuscript does not clearly define its objectives and hypotheses, particularly in lines 89 to 97. Once the objectives and hypotheses are clarified, please revise the conclusions to indicate whether the objectives were met and if the hypotheses were supported.
Author Response
Comments 1: Line 18: The phrase "To better understand the pattern of various testicular cells in spermatogenesis" serves as a vague and poorly defined objective. The authors need to provide a clear and specific objective.
Response 1: We have provided a more specific objective based on our research,thank you very much.
Comments 2: Lines 89 to 97: There is no need to include details about the materials and methods in this section, nor should findings or conclusions be presented here as it is not the appropriate location. The statement "To demonstrate the… and preservation of the race" should be removed. Instead, it is essential to clearly state the research objective and hypothesis.
Response 2: We have removed the statement "To demonstrate the… and breed preservation." and replaced the statement of research objective and hypothesis,thank you very much.
Comments 3: Lines 106 to 107. It is not clear how the testicles were obtained. Were the pigs castrated or were they killed and then got their testicles back? How many one-week-old and six-month-old pigs were used in the experiment?
Response 3: We have added specific descriptions in the Materials and Methods section2.1,thank you very much.
Comments 4: Materials and methods. Throughout the entire section, the authors must include a complete description of every one of the software, products, equipment, or commercial reagents used, to ensure the repeatability of the experiment. Please include information as follows: Trade name of the product, equipment, or reagent; name of the manufacturing company, place of manufacture.
Response 4: We have added specific information in the Materials and Methods section,thank you very much.
Comments 5: Results: The order of the sections is incorrect, which renders the manuscript confusing and difficult to follow. The authors should reorder the sections to precisely match the sequence presented in the Materials and Methods section.
Response 5: HE staining, immunohistochemistry and immunofluorescence analysis are all histological tests, so we grouped them together in the materials and methods 2.8. Now we have reorder the sections to match the results to the material and methods sections, thank you very much.
Comments 6: Lines 186 to 189: Please eliminate the redundant phrase “To observe the... cells were observed,” as this has already been described in the Materials and Methods.
Response 6: We have eliminated the redundant phrase, thank you very much.
Comments 7: Lines 259 to 260: Please remove the unnecessary statement “Somatic cells provide... and spermatogenesis.”
Response 7: We have removed the unnecessary statement, thank you very much.
Comments 8: Lines 289 to 290: Please delete the redundant text “Spermatogenesis is a... differentiation.”
Response 8: We have deleted the redundant text, thank you very much.
Comments 9: Lines 342 to 343: Please remove the statement “China has abundant... different regions,” since the study did not assess differences between pig breeds, making this content irrelevant.
Response 9: We have removed the statement “China has abundant... different regions”, thank you very much.
Comments 10: Lines 378 to 379. Please remove the text “The results may vary depending on the species, breed, and age”. The study did not evaluate differences between breeds of pigs, so the authors are assuming untested results.
Response 10: We have removed the text “The results may vary depending on the species, breed, and age”, thank you very much.
Comments 11: Conclusions (pp. 429-435): The authors present a thorough summary of the manuscript; however, the overall organization and direction are lacking. The manuscript does not clearly define its objectives and hypotheses, particularly in lines 89 to 97. Once the objectives and hypotheses are clarified, please revise the conclusions to indicate whether the objectives were met and if the hypotheses were supported.
Response 11: We have already revised the objectives and the conclusions, thank you very much.
Round 2
Reviewer 2 Report
Comments and Suggestions for Authors
In the revised version, all comments concerning the previous version were adequately addressed.